# Comparison of Regional Urban Water Pollutants Emission Standards and Determination of Factors Influencing Their Integration—A Case Study of the Biopharmaceutical Industry in the Yangtze River Delta Urban Agglomeration

Liping Cao [1,*], Xinyu Liu [1], Shuai Zhang [2] and Mingjie Lyu [3,*]

1   Institute of Ecology and Sustainable Development, Shanghai Academy of Social Sciences, Shanghai 200020, China; kingjim@sass.org.cn
2   College of Design and Creativity, Tongji University, Shanghai 200092, China; zhangshuaiboshi@tongji.edu.cn
3   Accounting College, Shanghai Lixin University of Accounting and Finance, Shanghai 201620, China
*   Correspondence: caoliping@sass.org.cn (L.C.); lmj86112@126.com (M.L.)

**Abstract:** Urban pharmaceutical industries are responsible for high intensity emissions of water pollutants. The regional water pollutant emission standards vary greatly throughout the Yangtze River Delta Urban Agglomeration (YRDUA) in China, which, to some extent, results in increased risks and hidden dangers to regional water environment safety, especially water quality. Under the national strategy of Yangtze River Delta ecological and green integration development, a unified standard of water pollutant emissions should be integrated into the integration development process, but differences between characteristic items, concentration limits, and conditions among four local standards of water pollutant emission have become a key influencing factor in their integration in industry and in green transformation. When comparing the water pollutant emission standards of the biopharmaceutical industry in three provinces and one municipality of the YRDUA, the factors influencing integration were determined and caused by the main differences in local water pollutant emission standards, namely, the race to the bottom of the biopharmaceutical industry, the inconsistency of environmental protection regulation law, and transboundary water pollution risks. From the perspective of urban water quality safety, we propose the following strategies for promoting the integration of water pollutant emission standards in the YRDUA: (1) increasing government funding for local water pollution governance and encouraging industries to adopt the third-party governance model for pollution control in the YRDUA; (2) unifying water pollutant emission standards and environmental law enforcement standards in the YRDUA with a mechanism involving shared economic responsibility; and (3) establishing a platform for sharing data and governance performance for the emission of water pollutants in the YRDUA.

**Keywords:** urban water; water pollutant; emission standards; integration; influencing factors

## 1. Introduction

Market-based approaches and command and control regulations are the two main tools that policy/decision makers use to implement environmental protection goals [1]. In China, command and control regulations like pollutant emission standards are more efficient than market-based approaches; however, in regions with similar local geographies, each city's water pollutant emission standards are in accordance with superior standards, which will increase the institutional cost for managing the regional water environment [2]. Moreover, this research wants to find the optimized path to decrease institutional costs by integrating regional environment regulations. In addition, earlier research has indicated that the water quality safety in the Yangtze River Delta Urban Agglomeration (YRDUA) is influenced by the south-to-north water transfer project from downstream of the Yangtze River, which

causes the estuary water to be too low in dry years and months, resulting in the urban water in the estuary area of the Yangtze River being too salty for human consumption [3,4]. According to the latest studies, from 2013 to 2017, the urban water quality situation in the YRDUA has continuously improved, but there are still three significant safety risks, namely, (a) the urban water consumption in this region is large, and the quality of water supplies is greatly affected by the amount and quality of incoming water from upstream of the Yangtze River, and there is seasonal variation in the risks associated with drinking the water due to its quality; (b) there are large amounts of urban wastewater emissions and ammonia nitrogen emissions in the water, and the cost of wastewater treatment is also substantial; (c) the annual average number of sudden water environment safety incidents between cities in this region is high, which results in uncertainty, due to safety risks to urban surface water quality downstream [5]. The wastewater emissions of the pharmaceutical industry in the YRDUA region are also now a potential factor contributing to the risks detailed in (a) and (b). There are more than a thousand pharmaceutical enterprises in the YRDUA, and in 2018, about 82.34 million m$^3$ of wastewater and 334 tons of ammonia nitrogen emitted by pharmaceutical manufacturing industries in this region, separately accounted for 2.6% and 3% of regional industrial emissions. From the perspective of urban water environment safety, the water pollutant emission standard, as a kind of environment regulation, is the bottom line to ensure that regional and urban water environmental quality are not further degraded [6]; however, even though three provinces and one municipality in the YRDUA have separately issued water pollutant emission standards to the pharmaceutical industry, and the quality of wastewater emitted by pharmaceutical enterprises should meet the water pollutant emission standards in provinces upstream by the upstream provinces, when this water flows to the area downstream, the water quality no longer meets the corresponding standards issued by the downstream provinces or municipalities because of the different environmental standards of the different areas [7]. As a result, the industrial chain layout in the YRDUA is unbalanced, and industries with excessive pollution emissions gather in places where there are low standards regarding the level of water pollutant emissions [8]. As a type of pharmaceutical industry related to human health, the biopharmaceutical industry is an emerging industry which exhibits clustering in the YRDUA, including biological engineering, fermentation, extraction, and preparation processes using organisms or biological processes for manufacturing drugs, with biomedical research and development institutes also included according to the water pollutant emissions standards of the biopharmaceutical industry in Shanghai Municipality, which subsequently form the corresponding five sub-industries of the biopharmaceutical industry. The fermentation pharmaceutical industry produces a higher chemical oxygen demand (COD) and more ammonia nitrogen water pollutants among the five sub-industries of the biopharmaceutical industry; the level of pollutant emissions in the biopharmaceutical process is huge [9] and the cost of wastewater treatment is relatively high, generally about 16 yuan (RMB)/t. "Yuan" is the unit of currency in China and "RMB" represents Ren Min Bi, which is the corresponding sign for this currency. The extraction pharmaceutical industry of biopharmaceuticals uses a large number of organic solvents, which are sometimes emitted in a disorganized manner. To an extent, they damage the regional urban water environment quality [10] and the cost of wastewater treatment in extraction pharmaceutical industry exceeds 10 yuan (RMB)/t. The amount of wastewater pollution that is emitted by the biological engineering industry and preparation industry is not huge, and their cost of wastewater treatment is considered low, generally less than 5 yuan (RMB)/t. The water pollutant emissions by biomedical research and development (R&D) institutes are characterized by complexity and diversity, and the emission concentrations of water pollutants and the cost of wastewater treatment are uncertain. The wastewater emission from the biopharmaceutical industry in the YRDUA has caused water quality risks and has resulted in pressure for the governance of pollution in the water environment [11]. The updated country emission standard (GB 37823-2019) for air pollutants of the pharmaceutical industry was issued in 2019 by the Ministry of Ecology and Environment of China, and an integrated standard for air pollutant emissions in the

YRDUA has been jointly established by the three provinces and one municipality of this region in 2020, but still hasn't been officially released; however, no unified standard for water pollutant emissions exists in the YRDUA because it has been too difficult to unify the complex items of water pollutants, and the accumulation of some water pollutants, such as mercury, sulfur and nitrogen, is beyond the capabilities of the monitoring system.

Following the abovementioned considerations, we take the water pollutant emission standards of biopharmaceutical industry in the YRDUA, for example, and use statistical and comparative analysis methods to compare water pollutant emission standards in four jurisdictions within the YRDUA to determine water environmental integration and coordination within the region; a comparative analysis has never been done in previous studies. Water environmental integration occurs under the request of the national strategy, the "Yangtze River Delta integration strategy" [12]. The result of water environmental integration is to improve the whole water quality in the YRDUA, which needs multi-government coordination to manage the environmental system within the region, such as unified standards, unified supervision, and unified law enforcement.

Moreover, through a comparison study on the differences between water pollutant emission standards in the three provinces and one municipality of the YRDUA, we know the importance of integrating urban water pollutant emission standards in promoting integration of regional eco-environmental quality. In addition, countermeasures and suggestions for the integration of water pollutant emission standards will be put forward. These are the policy implications of this paper.

## 2. Materials and Methods

### 2.1. Study Area

The YRDUA accounts for 11.7% of the national population and 2.14% of the total land area of China, and its output accounts for about 20% of total GDP [13,14]. As a region, it is one of the largest contributors to the Chinese economy, and it has the highest intensity of human development activities, consumption of various resources and energy, and emission intensity of pollutants. Prior to the release of the Development Plan of the YRDUA (2016–2030), provinces and cities coped with environmental management independently, with prominent overlaps, conflicts, and contradictions between the plans. Although the provinces and cities have a highly accepting attitude toward the integration of the supply of ecological products and environmental services and the integrated layout of environmental infrastructure construction in the YRDUA, it is difficult to achieve in-depth cooperation in terms of actual actions. This is due to the lack of an integrated scientific standard and guidance for environmental management, which restricts the coordinated development of ecological environmental governance and ecological security maintenance in the process of regional urbanization [14].

With the continuous evolution of regional urbanization, a series of ecological and environmental problems have aroused common concern (e.g., air and water pollution [15]). Surface runoff pollution emitted by industries increases the risk to urban drinking water quality, with serious eutrophication of Taihu Lake Basin.

The YRDUA is located in the lower reaches of the Yangtze River in Eastern China (Figure 1). According to the statistics and analysis of the China Pharmaceutical Industry Information Center, there are 33 pharmaceutical enterprises in the YRDUA that are also in the list of top 100 pharmaceutical enterprises in China, as of 2019 (Table 1) [16], and their total revenues account for 35% of the top 100 pharmaceutical enterprises. Among them, 14 are in Jiangsu Province, 11 are in Zhejiang Province, 7 are in Shanghai Municipality, and 1 is in Anhui Province. The Jiangsu Province has formed a "Traditional China pharmaceuticals urban agglomeration" with Taizhou city at the core; Nanjing city, Changzhou city, Wuxi city, Suzhou city, and Nantong city along the Yangtze River at the southern wing; and Lianyungang city and Xuzhou city forming the North Wing. This reflects the 'One Center with Two Wings' pharmaceutical industry development pattern. The top four pharmaceutical enterprises in the YRDUA are the Jiangsu Yangzijiang Pharmaceutical

Group, the Shanghai Pharmaceutical Group, the Shanghai Fosun Pharmaceutical Group, and Jiangsu Hengrui Pharmaceuticals. The main business of these four pharmaceutical companies is chemical pharmaceuticals, and their total revenue in 2019 was about RMB 111.4 billion, accounting for about 12% of the total income of the top 100 pharmaceutical enterprises and 16% of the pharmaceutical industry in the YRDUA. It is a low-concentration competitive market, indicating that Jiangsu Province, Zhejiang Province, and Shanghai Municipality are the main sites of the pharmaceutical industry in the YRDUA, which are well-matched in strength.

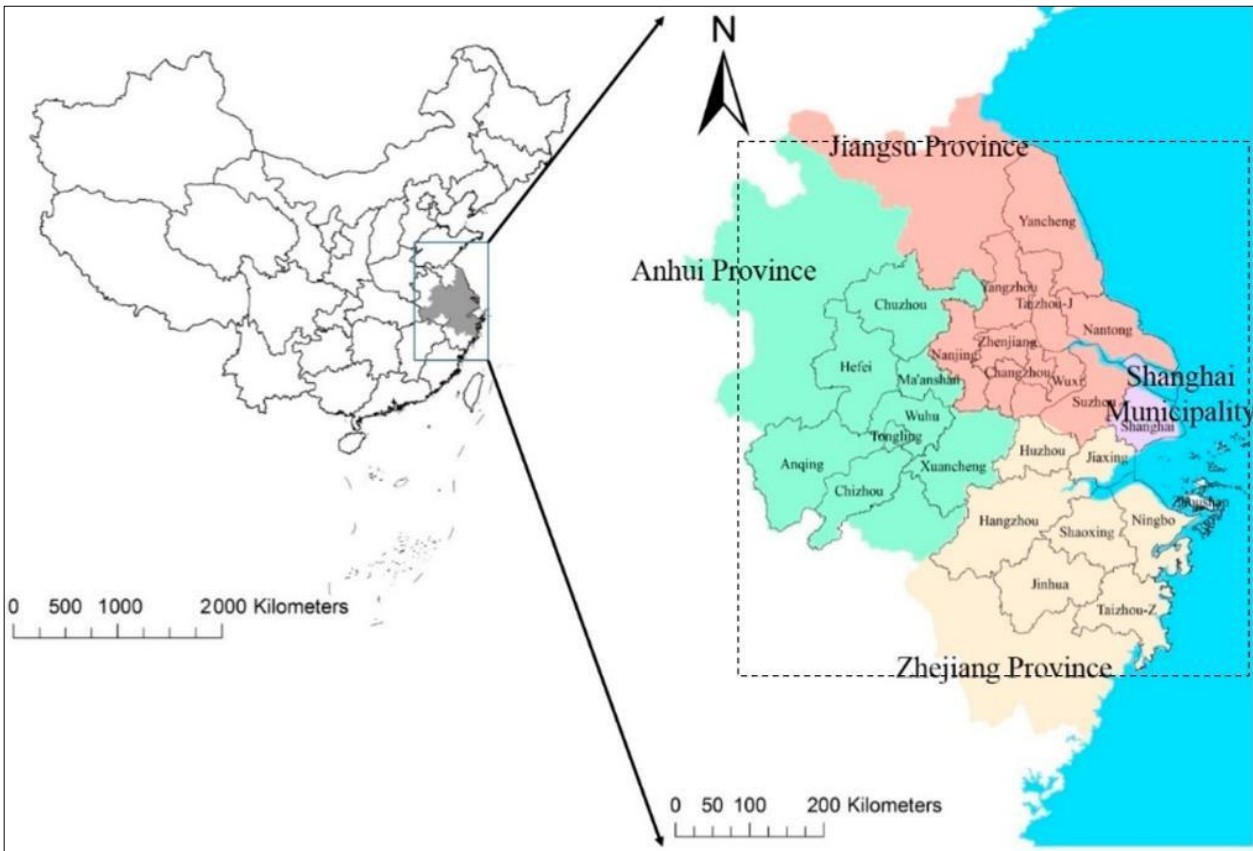

**Figure 1.** Location of the Yangtze River Delta Urban Agglomeration.

### 2.2. Methods

This paper uses two kinds of methods for finding the determination of factors influencing the integration of regional urban water pollutant emission standards.

One method is statistical and comparative analysis, which is used to calculate and compare the status quo of the wastewater emission efficiency of the pharmaceutical industry in the YRDUA with Formula (1), and comparative analysis is also used to compare regional urban water pollutant emission standards in the YRDUA. During the comparison anaylsis of the status quo of wastewater emission efficiency, the amount and percentage of the wastewater emission or water pollutant emission is the scale comparision, and the output value per wastewater of water pollutant emission is the efficiency comparison, which follows Formula (1):

$$\text{wastewater emission efficiency} = \frac{\text{the amount of output value of pharmaceutical industry}}{\text{the amount of wastewater or water pollutant emission}} \qquad (1)$$

In Formula (1), the amount of output value from the pharmaceutical industry and the amount of wastewater or water pollutant emission both came from the local statistical yearbook (2019 and 2020).

**Table 1.** The ranking, location, and classification of pharmaceutical enterprises in the YRDUA entering the top 100 pharmaceutical enterprises in China, 2019 [16].

| Ranking According to Income | Enterprises | Location in the YRDUA | Main Business Classification |
|---|---|---|---|
| 1 | Yangzijiang Pharmaceutical Group Co., Ltd. | Jiangsu Province | Chemical pharmaceuticals |
| 6 | Shanghai Pharmaceutical (Group) Co., Ltd. | Shanghai Municipality | Chemical pharmaceuticals |
| 7 | Shanghai Fosun Pharma (Group) Co., Ltd. | Shanghai Municipality | Biopharmaceuticals |
| 9 | Jiangsu Hengrui Pharmaceutical Co., Ltd. | Jiangsu Province | Chemical pharmaceuticals |
| 16 | Zhengda Tianqing Pharmaceutical Group Co., Ltd. | Jiangsu Province | R&D pharmaceuticals |
| 17 | Shanghai Roche Pharmaceutical Co., Ltd. | Shanghai Municipality | Biopharmaceuticals |
| 18 | AstraZeneca Limited | Jiangsu Province | Preparation's pharmaceuticals |
| 23 | Sanofi (Hangzhou) Pharmaceutical Co., Ltd. | Zhejiang Province | Biopharmaceuticals |
| 28 | Hangzhou Mercado Pharmaceutical Co., Ltd. | Zhejiang Province | Biopharmaceuticals |
| 32 | Jiangsu Howson Pharmaceutical Group Co., Ltd. | Jiangsu Province | R&D pharmaceuticals |
| 34 | Jiangsu Jichuan Holdings Group Co., Ltd. | Jiangsu Province | Traditional China pharmaceuticals |
| 37 | Xinhe Cheng Holding Group Co., Ltd. | Zhejiang Province | Chemical pharmaceuticals |
| 41 | Jiangsu Kangyuan Group Co., Ltd. | Jiangsu Province | Traditional China pharmaceuticals |
| 42 | Connbe Group Co., Ltd. | Zhejiang Province | Chemical pharmaceuticals |
| 46 | Plo Pharmaceutical Co., Ltd. | Zhejiang Province | Chemical pharmaceuticals |
| 47 | Huali Pharmaceutical Group Co., Ltd. | Zhejiang Province | Chemical pharmaceuticals |
| 50 | Zhejiang Haizheng Pharmaceutical Co., Ltd. | Zhejiang Province | Chemical pharmaceuticals |
| 51 | Zhejiang Huahai Pharmaceutical Co., Ltd. | Zhejiang Province | Chemical pharmaceuticals |
| 54 | Nanjing Xiansheng Dongyuan Pharmaceutical Co., Ltd. | Jiangsu Province | Chemical pharmaceuticals |
| 55 | Sino-American Shanghai Squibb Pharmaceutical Co., Ltd. | Shanghai Municipality | Chemical pharmaceuticals |
| 58 | Zhejiang Pharmaceutical Co., Ltd. | Zhejiang Province | Chemical pharmaceuticals |
| 59 | Jiangsu Osaikang Pharmaceutical Co., Ltd. | Jiangsu Province | R&D pharmaceuticals |
| 70 | Jingxin Holdings Group Co., Ltd. | Zhejiang Province | Chemical pharmaceuticals |
| 71 | Eisai (China) Investment Co., Ltd. | Jiangsu Province | Traditional China pharmaceuticals |
| 72 | Shanghai Acebright Pharmaceutical Group Co., Ltd. | Shanghai Municipality | R&D pharmaceuticals |
| 74 | Zhejiang Xianxian Pharmaceutical Co., Ltd. | Zhejiang Province | Chemical pharmaceuticals |
| 78 | Jiangsu Suzhong Traditional Chinese Pharmaceutical Group Co., Ltd. | Jiangsu Province | Traditional China pharmaceuticals |
| 80 | Jiangsu Yabang Pharmaceutical Group Co., Ltd. | Jiangsu Province | Chemical pharmaceuticals |
| 81 | Shanghai Bollinger Ingleham Pharmaceutical Co., Ltd. | Shanghai Municipality | Chemical pharmaceuticals |
| 84 | Wyeth Pharmaceuticals Co., Ltd. | Jiangsu Province | Biopharmaceuticals |
| 90 | Baxter (China) Investment Co., Ltd. | Shanghai Municipality | Device pharmaceuticals |
| 95 | Anhui Fengyuan Group Co., Ltd. | Anhui Province | Chemical pharmaceuticals |
| 100 | Jiangsu Enhua Pharmaceutical Co., Ltd. | Jiangsu Province | Chemical pharmaceuticals |

The other method uses a case study. As important environmental regulations for the wastewater emissions of the pharmaceutical manufacturing industry in the YRDUA, the national and local water pollutant emission standards of the pharmaceutical industry are the bottom line for water pollution prevention and control in the pharmaceutical manufacturing industry in the YRDUA. Among them, the water pollutant emission standards of the maximized chemical synthesis pharmaceutical industry and the smaller traditional Chinese medicine pharmaceutical industry in the YRDUA are basically consistent with the strictest national industry standards, except in the case of the Anhui Province, who use the national standard of water pollutants emission for the pharmaceutical industry (GB21904-2008); and GB21904-2008 is established for Tai Lake Basin. Moreover, the three provinces and one municipality in the YRDUA have formulated local standards for water pollutant emissions of the biopharmaceutical industry and its sub-industries, resulting in differentiated management of water quality throughout the region. This paper takes the biopharmaceutical industry as an example to compare the local water pollutant emission standards of the biopharmaceutical industry and its sub-industries for the three provinces and one municipality in the YRDUA, in terms of characteristic pollutant items, emission conditions, emission concentration limits, and industry types which come from the local standards of the biopharmaceutical industry. The pollutant emission standard for the biopharmaceutical industry of Shanghai Municipality is DB31/373-2010, of Zhejiang Province it is DB33/923-2014, of Jiangsu Province it is DB32/3560-2019, and of Anhui Province it is DB34/2710-2016, but Anhui Province mainly uses national standard GB8978-1996. Based on the difficulties of analyzing the integration of the biopharmaceutical industry's water pollutant emission standards in the YRDUA, we also use a sustainable policy analysis method, which is the Pressure-State-Response (PSR) model. This was developed by an organization for economic co-operation and development (OECD) and the United Nations Environment Programme (UNEP) in the 1980s and 1990s to reflect the interaction between human behaviors and the environment [17]; therefore, suggestions for the integration of water pollutant emission standards are put forward.

## 3. Results and Discussion

### 3.1. Results

3.1.1. Comparison of Wastewater Emission Efficiency of the Pharmaceutical Industry in the YRDUA

Statistically, there were about 6000 pharmaceutical enterprises in China in 2017, and the total output value ratio of the pharmaceutical industry accounted for about 2.1% of the national total industrial output value, whereas the total ratio of pollutant emissions reached about 6% of pharmaceutical wastewater emissions, and COD emissions account for about 3% of the national industry, which indicates that they have become the major polluters [18]. Water pollutant emissions from China's pharmaceutical industry includes three categories: one due to production processes, one due to backward manufacturing techniques, and one due to poor management. The three categories of water pollution have difficult and complex characteristics for governance, and a large amount of wastewater and similar water pollutants are produced in drug substance and fermentation pharmaceutical production processes. The treatment processing cost of this is higher, especially given the loads from the chemical crude drugs manufacturing industry, which represents the largest amount of pollution in the pharmaceutical manufacturing industry, as their wastewater emissions account for about 80% of the whole pharmaceutical manufacturing industry; therefore, monitoring pollution sources has become the key objective of the Environmental Protection Department of China. In January 2020, the Committee of Industry and Information Technology, the Ministry of Ecological Environment, the National Health Commission, and the National Medical Products Administration jointly issued guidelines to promote the development of active pharmaceutical ingredients for a greener industry. The Ministry of Ecological Environment also immediately issued three kinds of applications for industrial pollutant emission permits and technical specifications, including chemical pharmaceutical

preparation manufacturing, biological pharmaceutical products manufacturing, and Chinese patent medicine production, which will contribute to the high-quality development and green transformation of China's pharmaceutical manufacturing industry.

The three provinces and one municipality in the YRDUA have also stepped up efforts to control water pollution in the pharmaceutical manufacturing industry. According to statistics, the level of pharmaceutical manufacturing wastewater emissions in the YRDUA in 2018 is down more than 4% from the previous year. Among them, COD and ammonia nitrogen, which are the main water pollutants resulting from pharmaceutical and biopharmaceutical chemical synthesis, were about 6100 tons and 334 tons, respectively, in 2018 (Figure 2). This accounts for about 3.4% and 3% of the industrial pollutant emissions in the YRDUA, down 21% and 32%, respectively, from the previous year. In terms of geographical distribution and emission reduction contribution, Jiangsu Province and Zhejiang Province had the highest amount and percentage of pharmaceutical wastewater and water pollutant emissions of the three provinces and one municipality in the YRDUA, in 2018, with a double-digit rate of decline in terms of their contribution to the regional pharmaceutical industry's reduction of water pollution. The growth in the proportion of pharmaceutical wastewater and water pollutant emissions in Shanghai Municipality also dropped by double-digits, driven by the municipality's refined management. Although the amount of pharmaceutical wastewater and water pollutants in the Anhui Province is the lowest in the YRDUA, it holds a large number of pharmaceutical enterprises, and is the second highest after Jiangsu Province, which has the largest number of pharmaceutical enterprises with a relatively scattered distribution, and thus, it contributes least to the reduction of water pollutants in the regional pharmaceutical industry.

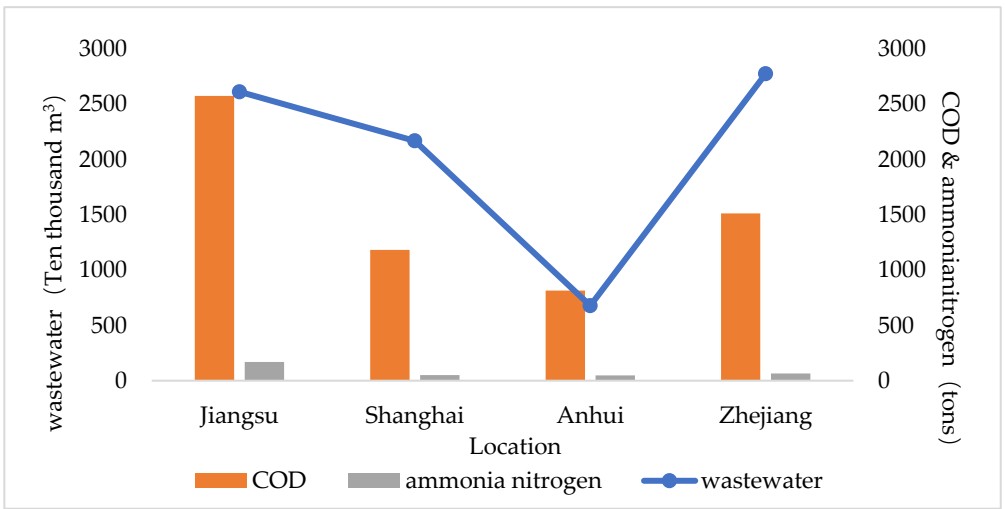

**Figure 2.** Comparison of the amount of wastewater and water pollutant emissions from the pharmaceutical industry in the three provinces and one municipality of the YRDUA in 2018.

From the perspective of wastewater emission efficiency in the pharmaceutical manufacturing industry, the output value per wastewater emission unit of the pharmaceutical industry in 2017 and 2018 for the YRDUA provinces and municipality was higher than the national average level of 0.4 thousand m$^3$/yuan (Figure 3). The output value per COD emission unit of the pharmaceutical industry in the YRDUA provinces and municipality in 2018 is similar, at about one hundred million ton/yuan (Figure 4). The output values per ammonia nitrogen emission unit of the pharmaceutical industry in Shanghai Municipality and Zhejiang Province in 2018 are higher, whereas the output value per ammonia nitrogen emission unit of the pharmaceutical industry in the Jiangsu Province and Anhui Province is relatively low (Figure 5). It can be seen that the output value per wastewater and water pollutant emission unit of the pharmaceutical industry in the three provinces and one municipality of the YRDUA vary greatly; however, the water source of Shanghai Municipality

originates from the main river channels downstream of the YRDUA, such as the Huangpu River and Suzhou River in Shanghai Municipality, with the two neighboring provinces of Jiangsu and Zhejiang being upstream. The wastewater emissions of the pharmaceutical manufacturing industry upstream of the main river channels in the Jiangsu Province and Zhejiang Province, especially for chemical synthesis, threatens the water quality and water source security in Shanghai Municipality.

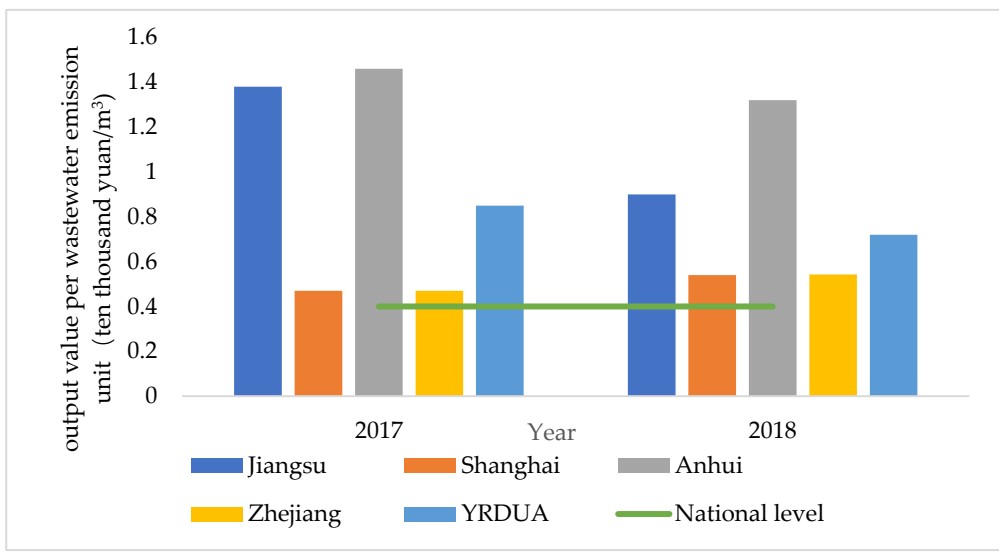

**Figure 3.** Comparison of the output value per wastewater emission unit of the pharmaceutical industry in the YRDUA provinces and municipality in 2017 and 2018.

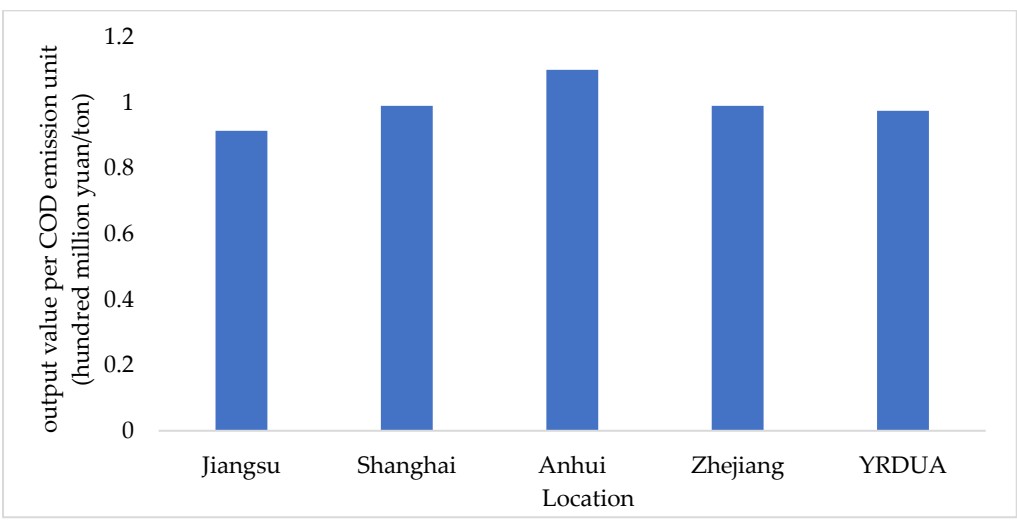

**Figure 4.** Comparison of the output value per COD emission unit of the pharmaceutical industry in the YRDUA provinces and municipality in 2018.

3.1.2. Comparison of Local Standards of Water Pollutant Emission in the YRDUA: A Case Study of the Biopharmaceutical Industry

The Characteristic Items, Concentration Limits, and Conditions for the Emission of Water Pollutants Differ According to the Standards for the Biopharmaceutical Industry and Sub-Industries

First of all, the characteristic items, concentration limits, and conditions for water pollutants of Category I or Category II differ according to the standards of the biopharmaceutical industry and sub-industries. For example, the water pollutant emission standard of the biopharmaceutical industry in Shanghai Municipality states that total selenium is

listed as a characteristic Category I water pollutant item, but the Jiangsu Province standard (DB32/3560-2019) lists it as a Category II item. Moreover, total selenium is not considered a characteristic water pollutant item in the corresponding Zhejiang Province standard (Table 2).

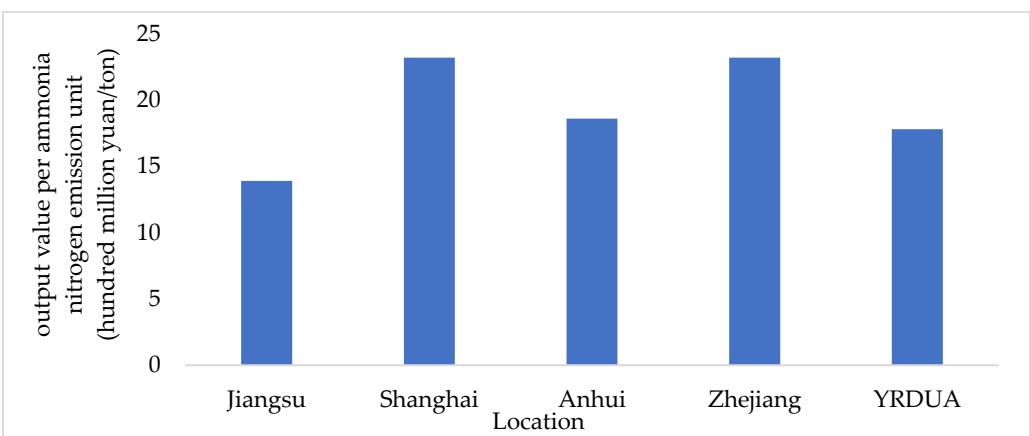

**Figure 5.** Comparison of the output value per ammonia nitrogen emission unit of the pharmaceutical industry in the YRDUA provinces and municipality in 2018.

Second, Table 1 shows that the water pollutant emission conditions for Category I and Category II are not uniform among the various local standards in terms of the water emission sources. (1) Water and air pollutant emission standard for the biopharmaceutical industry of Shanghai Municipality (DB31/373 2010) distinguishes, in detail, the emission concentration limits for water pollutants of two types of pollution sources corresponding to different receiving water bodies and new or old sewage emission enterprises. (2) Water and air pollutant emission standards for the biopharmaceutical industry of Jiangsu Province and Zhejiang Province (DB32/3560-2019, DB33/923-2014) also distinguish different receiving water bodies and pollution sources from new or old emission enterprises in terms of the concentration limits for the emission of Category II water pollutants; however, although the Jiangsu Province standard distinguishes between new and old enterprises in terms of the concentration limit for the emission of Category I water pollutants, it does not distinguish between receiving water bodies. In addition, the concentration limit for the emission of Category I water pollutants, as per the Zhejiang Province standard, distinguishes receiving water bodies, but does not distinguish new or old enterprises. (3) There is no local standard for water pollutant emissions of the biopharmaceutical industry of the entire Anhui Province, which is located upstream of the YRDUA. Moreover, Anhui Province has only issued a water pollutant emission concentration limit standard for major water pollutants from urban sewage treatment plants (DB34/2710-2016). This standard is especially for industrial enterprises in the Chaohu Lake Basin, which is part of the administrative division of Anhui Province. But this standard does not distinguish between the conditions for emission of Category I and Category II water pollutants, such as whether the source of pollutants is from new or old sewage emission enterprises (Table 3). Although Anhui Province have issued the air pollutants emission standard for the pharmaceutical industry in 2021 (DB34/310005-2021), it has still not issued the water pollutants emission standard for the pharmaceutical industry.

**Table 2.** Comparison of the characteristic items, conditions, and concentration limits for the emission of Category I water pollutants for the biopharmaceutical industry in the YRDUA (unit of water pollutant emission concentration limit: mg/L).

| Comparison Items | | Shanghai Municipality | | | | Zhejiang Province | | Jiangsu Province | | Anhui Province |
|---|---|---|---|---|---|---|---|---|---|---|
| Standard number in different area | | DB31373-2010 | | | | DB33023-2014 | | DB32/3560-2019 | | GB8978-1996 |
| Time of standard execution | | 1 January 2010 | | | | 1 February 2015 | | 1 April 2019 | 1 April 2021 | 1 January 1998 |
| Emission conditions | Emission source | Existing (all) | Existing (Built after 1 January 2010) | New and Existing (with EIA documents from 1 February 2009, to 30 June 2010) | Existing (all) and New | Existing (all) and New | | New | Existing (all) | Existing (all) and New |
| | Receiving waters/facilities | Special Protected waters | General waters | | Terminal sewage treatment facilities | Special Protected waters | General waters and Terminal sewage treatment facilities | Not specified | | Not specified |
| Emission characteristic items and concentration limit (mg/L) | Total mercury (by Hg) | 0.005 | 0.01 | 0.005 | 0.01 | 0.005 | 0.01 | 0.05 | | 0.05 |
| | Alkyl mercury (by Hg) | ND | ND | ND | ND | ND | ND | 0.0003 | | ND |
| | Total Cadmium (by Cd) | 0.01 | 0.1 | 0.01 | 0.1 | 0.1 | 0.1 | 0.1 | | 0.1 |
| | Total chromium (by Cr) | 0.15 | 1.5 | 0.15 | 1.5 | 0.15 | 1.5 | 0.15 | | 1.5 |
| | Hexavalent chromium (by $Cr^{6+}$) | 0.05 | 0.1 | 0.05 | 0.1 | 0.05 | 0.1 | 0.05 | | 0.5 |
| | Total arsenic (by As) | 0.05 | 0.1 | 0.05 | 0.1 | 0.05 | 0.1 | 0.05 | | 0.5 |
| | Total selenium (by SE) | 0.1 | 0.1 | 0.1 | 0.1 | NA | NA | 0.1 (Category II) | | NA |

Note: ND = not detected; NA = not applicable; Existing means existing pollution source; New means new pollution source; EIA means environmental impact assessment.

**Table 3.** Comparison of conditions for water pollutant emissions of the biopharmaceutical industry in the three provinces and one municipality of the YRDUA.

| Location | Whether It Distinguishes Different Receiving Water Bodies | | Whether It Distinguishes between Old and New Sources of Pollution | |
|---|---|---|---|---|
| | Category I Water Pollutants | Category II Water Pollutants | Category I Water Pollutants | Category II Water Pollutants |
| Shanghai Municipality | Yes | Yes | Yes | Yes |
| Jiangsu Province | No | Yes | Yes | Yes |
| Zhejiang Province | Yes | Yes | No | Yes |
| Anhui Province | No | No | No | No |

The Limits of Water Pollutant Emission Concentrations in the Jiangsu Province Standard Are Ahead of Zhejiang Province, Shanghai Municipality, and Anhui Province Standards

The water pollutant emission standard of the biological pharmaceutical industry in Jiangsu Province of the YRDUA is the latest local standard. Its standard not only differs from Zhejiang Province, Shanghai Municipality, and Anhui Province standards in terms of the classification and emission conditions of characteristic water pollutant items, but also in terms of the limits of water pollutant emission concentrations.

First of all, in terms of the concentration limit of the emission of Category I water pollutants by the biopharmaceutical industry, as specified in the standard, the emission concentration limit of Jiangsu Province standard (except in the case of alkyl mercury) is stricter than that of other standards in the YRDUA and the national comprehensive sewage emission standard (GB8978-1996). In addition, the adopted concentration limits for the emission of other Category I water pollutants represent the strictest limits of all standards for water bodies under special protection, without distinguishing between receiving water bodies and pollutant emission sources (i.e., whether they were new or old). For water pollutant emissions from old sewage emission enterprises built before 1 July 2010, that run into general environmental water bodies, and are from all enterprises emitting water pollutants into urban or industrial drainage systems with terminal sewage treatment facilities, the emission concentration limits for Category I water pollutants are lower in Shanghai Municipality than in Jiangsu Province standard; however, for the water pollutant emissions from new sewage emission enterprises built after 1 July 2010, with the environment impact assessment (EIA) occurring between 1 February 2009 to 30 June 2010, their water pollutant emissions running into general environmental water bodies, and the concentration limits for the emission of Category I water pollutants from all enterprises running into specially protected water bodies, they are the same as for Jiangsu Province standard. After a certain period of time, these differences will lead to differences in the concentration of water pollutants emitted by old and new sewage emission enterprises in the biopharmaceutical industry in Shanghai Municipality, and it could cause inequity due to the differences in the treatment costs between the old and new sewage emission enterprises. The water pollutant concentration limit stipulated by Zhejiang Province standard for all enterprises, with regard to emissions in specially protected waters, is consistent with that of the Category I water pollutant limit in Jiangsu Province standard. The concentration limit for all enterprises, regarding emissions into general environmental water bodies and into drainage systems in towns or industrial parks with terminal sewage treatment facilities, is lower in Zhejiang Province standard than in Jiangsu standard. The concentration limit for the emission of Category I water pollutants is lower in the standard for Anhui Province (Chaohu Lake Basin) than the national comprehensive sewage emission standard implemented in other places in the Yangtze River Delta; therefore, in terms of the concentration limits for the emission of Category I water pollutants in the different standards of the biological pharmaceutical industry in the YRDUA, the different degrees of strictness in the local standards (Table 4) can potentially greatly endanger environmental water security in Anhui Province.

**Table 4.** Comparison of the degrees of strictness of local water pollutant emission concentration limit standards of Category I for the biopharmaceutical industry in the YRDUA.

| Standard of Classification | New Polluters | | | Existing Polluters | | |
|---|---|---|---|---|---|---|
| Location | Emission into Special Protected Waters | Emission to General Environmental Water Bodies | Emission to Drainage Systems in Towns or Industrial Parks Where Terminal Sewage Treatment Facilities Are Installed | Emission into Special Protected Waters | Emission to General Environmental Water Bodies | Emission to Drainage Systems in Towns or Industrial Parks Where Terminal Sewage Treatment Facilities Are Installed |
| Jiangsu Province | Strict | Strict | Strict | Strict | Strict | Strict |
| Shanghai Municipality | Strict | Strict | Relaxed | Strict | Relaxed | Relaxed |
| Zhejiang Province | Strict | Relaxed | Relaxed | Strict | Relaxed | Relaxed |
| Anhui Province (Chaohu Lake Basin) | Very relaxed | Very relaxed | Very relaxed | Very relaxed | Very relaxed | Very relaxed |

Second, in addition to Zhejiang Province, Anhui Province standard specifies comprehensive emission concentration limits for Category II water pollutants, in accordance with the biopharmaceutical industry standards. In addition, the emission concentration limits for Category II water pollutants in Jiangsu Province and Shanghai Municipality standards are subdivided into biological engineering, fermentation, extraction and preparation, and research and development, according to five sub-industry standards. Among them, the water pollutant concentration emission limits in the standards of sub-industries, such as bioengineering, fermentation, and extraction, are highly similar. As for the concentration limits for water pollutant emission into general environmental water bodies, the limits in Jiangsu Province standard are basically the same as those of Zhejiang Province and Anhui Province standards, which, in Shanghai Municipality standard, are within the pollutant concentration limits for new and old enterprises. The concentration limits regarding emission of water pollutants into specially protected water bodies is basically the same in Shanghai Municipality and Zhejiang Province standards, which are stricter than those of the Anhui Province standard.

Third, in terms of the standard for the biopharmaceutical industry, the emission of Category II water pollutants in the preparation sub-industry in the YRDUA, Jiangsu Province, and Shanghai Municipality have formulated local standards, and Zhejiang Province has adopted the national industry standard (GB21907-2008). Although the emission concentration limits of the four types of water pollutants, namely COD, ammonia nitrogen, total nitrogen, and total phosphorus, in Anhui Province standard (Chaohu Lake Basin) are strict, the emission concentration limits for other water pollutants are, however, simply set to correspond with the first-level standards of the National Comprehensive Sewage Emission Standard (GB8978-1996), which are less strict than the local standards. Among them, the partial concentration limit for Category II water pollutants, with respect to the drainage system of towns or industrial parks with terminal sewage treatment facilities, is stricter in Jiangsu Province standard than the corresponding Shanghai Municipality standard and the national industry standard, which is also implemented by Zhejiang Province, and is less strict than the corresponding standard of the Anhui Province Chaohu Lake Watershed. The partial concentration limit for the emission of Category II water pollutants into general environmental water bodies for old sewage emission enterprises built before 1 July 2010, is stricter in Jiangsu Province standard than in Shanghai Municipality standard, and the corresponding standards for Zhejiang Province and Anhui Province (Chaohu Lake Watershed). It is also more relaxed than the corresponding limits for new sewage enterprises built after 1 July 2010, according to Shanghai Municipality standard. The partial concentration limit for the emission of Category II water pollutants into a water body under special

protection are basically the same in both Jiangsu Province and Shanghai Municipality standards, which are stricter than the corresponding national industry standards executed by Zhejiang Province and the corresponding standard in the Chaohu Lake Watershed of Anhui Province.

Jiangsu Province standard adopted phased unified emission concentration limits for all water pollutants, regarding emission via new or old sewage enterprises; that is, after a period of time, both the old and new sewage enterprises will adopt the strictest concentration limits for the emission of water pollutants. This simplifies the process, as it removes the differences specified for old and new sewage enterprises in the biological pharmaceutical industry; therefore, it can speed up unification of the water pollutant emission concentration limit standards for the biopharmaceutical industry in Jiangsu Province.

Shanghai Municipality Is the Leader in Issuing a Water Pollutant Emission Standard for Biomedical R&D Institutes, but Jiangsu Province Has the Standard with the Strictest Limits

Biomedical R&D institutes are indispensable for global innovation. China has not yet issued a national industry standard for the emission of water pollutants from biomedical research institutes; thus it lags behind the United States, which has issued such standards for the pharmaceutical industry under the guidance of the Water Pollutant Emission Standards System of the Clean Water Act (the first version was established in 1976, and the 2003 version represents the most recent update) [19]. Shanghai Municipality is the leader in issuing water pollutant emission standards for biomedical R&D institutes, and its emission conditions for characteristic water pollutant items are differentiated, similarly to those of the four other sub-industries. Jiangsu Province followed Shanghai Municipality in issuing limits in its standard for water pollutant emissions via biomedical R&D institutes. Zhejiang Province and Anhui Province have not issued local standards for such water pollutant emission limits and have adopted the national comprehensive sewage emission standard to limit the concentration of water pollutants in emissions by biomedical R&D institutes.

The concentration limit for the emission of Category II water pollutants by biomedical R&D institutes into the drainage system of towns or industrial parks with terminal sewage treatment facilities is stricter in Jiangsu Province standard than in Shanghai Municipality standard, and that of the national comprehensive sewage emission standard implemented by Zhejiang Province and Anhui Province. The concentration limit in Jiangsu Province for the emission of Category II water pollutants by biomedical R&D institutes directly into a general environmental water body still falls between the water pollutant emission concentration limits for new and old sewage emission enterprises in Shanghai Municipality standard and is stricter than the comprehensive national sewage emission standard implemented by Zhejiang Province and Anhui Province. Regarding the concentration limit for the emission of Category II water pollutants by biomedical R&D institutes directly into specially protected waters, in addition to the emission limits for total cyanide compounds, they are less strict in Jiangsu Province standard than in Shanghai Municipality standard, and the emission concentration limit for 1,2-dichloroethane is stricter in Jiangsu Province standard than in Shanghai Municipality standard. On the other hand, concentration limits for the emission of other Category II water pollutants by biomedical R&D institutes are the same in the standards for Jiangsu Province and Shanghai Municipality, and are stricter than those of Zhejiang Province and Anhui Province, which have implemented the national sewage comprehensive emission standard. Moreover, due to the lack of national standards for biomedical R&D institutes, the national comprehensive sewage emission standards implemented by Zhejiang Province and Anhui Province lack many characteristic items that are applicable to the emission of Category II water pollutants by biomedical R&D institutes.

### 3.2. Discussion

The biopharmaceutical industry in the YRDUA is a typical industry with inconsistencies in the standards for the emission of water pollutants, which is the state (S) of water quality management in the YRDUA. Taking the biopharmaceutical industry as an example,

three challenges create a pressure (P), with regard to unifying four local standards of water pollutant emission in the YRDUA: the demand for economic development; the different regulation scales of the different pollutant emission standards; and the lack of environmental data, which themselves are also subject to three influencing factors, namely that the differing standards for the emission of water pollutants in the YRDUA directly affect the spatial layout of the biopharmaceutical industry chain in the YRDUA, the effectiveness of the regional joint prevention and control mechanism, and the guarantee of water security in the YRDUA.

### 3.2.1. Competition to the Bottom of the Industry Is Not Conducive to the Optimization of the Spatial Layout of the Industrial Chain

Taking the biological pharmaceutical industry in the YRDUA as an example, some places with lower local water pollutant emission concentration limits, as specified in the standard for the biological pharmaceutical industry, will attract biopharmaceutical enterprises with higher water pollutant emissions. For example, fermentation pharmaceutical enterprises or bioengineering enterprises containing fermentation technology, will gather together in some local sites where economic development by pharmaceuticals is needed, and some localities will also attract biopharmaceutical enterprises based on tax incentives and talent support, even given their high pollution at the expense of the environment; therefore, the spatial disequilibrium has appeared in the allocation of capital, talent, and technology in the biological pharmaceutical industry of the YRDUA.

In the biopharmaceutical industry chain, biopharmaceutical R&D institutions belong to scientific and technological innovation enterprises, which make large investments in the early stages of the biopharmaceutical R&D process and throughout the long cycle of the entire pharmaceutical process, requiring a large amount of capital investment and the support of high-tech talent. At the same time, the price of new biological drugs for clinical application is high, but the market for domestic expansion is limited; therefore, in the YRDUA, enterprises within the first stage of biopharmaceutical R&D are more concentrated in southern Jiangsu Province, Shanghai Municipality, Zhejiang Province, and other economically developed cities along the coast, as they are places with ideal R&D systems, developed financial services, and they have high levels of pollution governance. For example, Zhangjiang Pharmaceutical Valley in Shanghai Municipality, which has the highest density of highly educated talent in the YRDUA and a complete R&D system, has attracted Novartis, Pfizer, Roche, GSK, AstraZeneca, Eli Lilly, AbbVie, Amgen, and eight other top global biomedical R&D enterprises, in addition to Shanghai Pharmaceuticals, Fosun Pharmaceutical Co., Yangzijiang Pharmaceutical Group, Hausen Pharmaceutical Inc., and other representatives of the top 100 domestic biomedical enterprises; however, the biological engineering pharmaceutical industry in the YRDUA, which includes the fermentation pharmaceutical process, will also emit a large volume of pollutants, and a lot of money will be spent on pollution governance due to its similarity to the chemical synthesis pharmaceutical industry in terms of wastewater emission. Most are concentrated in central Jiangsu Province, northern Jiangsu Province, Shanghai Municipality, and Zhejiang Province. The biological extraction preparation processes of the pharmaceutical industry have lower levels of water pollutant emission, with lower pollution governance costs; however, with the continuous improvement in pharmaceutical technology, the profits from this industry are also gradually decreasing, and thus, pharmaceutical enterprises for biological extraction preparations in the YRDUA are mainly distributed in Jiangsu Province and Zhejiang Province, which have a high level of pharmaceutical technology.

In general, the whole industry chain of the biopharmaceutical industry is mainly concentrated in Jiangsu Province and Zhejiang Province. The main reason is that Zhejiang Province has invested in high pollution governance to attract high-tech biopharmaceutical enterprises to settle in; however, the timing of the execution of the water pollutant standard in Jiangsu Province is later than for other regions, and the implementation of the highest standard by existing enterprises has been delayed to April 2021. This period of delay

represents a window of opportunity and is an important factor that may drive biological pharmaceutical enterprises with high pollution emissions to aggregate in the Jiangsu Province, so in a short time, Jiangsu Province may become a zone for the water pollutant emissions of biological pharmaceutical industries in the YRDUA.

### 3.2.2. Environmental Command and Control Regulations like Standards of Supervision and Law Enforcement Are Difficult to Be Consistent

The regional standards for the emission of water pollutants by the biological pharmaceutical industry are not unified, which also results in lack of unity regarding the management of such emissions by biopharmaceutical enterprises in the YRDUA. In the three provinces and one municipality in the YRDUA, especially in trans-boundary ecological environment spaces such as the Taipu River, the trans-boundary small watershed located in the Wujiang district of Jiangsu Province, and the Qingpu district of Shanghai Municipality, the efficiency of joint prevention and the governance mechanism of daily water pollution is low, which creates opportunities for enterprises to emit pollutants illegally and excessively, with the opportunity to seep into surrounding areas. For example, in 2017, during the first round of central environmental protection supervision, for five of the ten urban sewage treatment plants in Qingpu, in Shanghai Municipality, located in the YRDUA cross-border area, the of heavy metal emission levels in sludge were found to have exceeded the levels specified in the standard. These plants were located in Huaxin town, Baihe town, Xujing town, and Liantang town in Qingpu, in Shanghai Municipality. The main reason the levels were exceeded in the sludge of the Huaxin town sewage treatment plant is that the upstream heavy metal emission enterprises illegally and excessively emitted pollutants. In the same year, the Environmental Protection Bureau of Qingpu District inspected 346 enterprises in the district, and the investigation resulted in the identification of five enterprises being found to have committed serious and illegal pollutant emission, and they were fined RMB 1.64 million. Approximately eight to nine enterprises that were found to have bad emission behaviors were closed and their cases were transferred to the judicial department for serious environmental pollution [20]. The local environmental protection administrative departments in the YRDUA can take effective measures to prohibit local enterprises from stealing and excessively emitting pollutants; however, for the handling of illegal emissions in the trans-boundary eco-environmental space in the YRDUA, there are difficulties associated with obtaining sufficient evidence, handling the standards in unified terms, and case handling inconveniences. Especially when law enforcement is involved in cross-administrative environmental protection, the supervising departments of the three provinces and one municipality in the YRDUA only recognize the evidence materials collected by the local departments themselves [21]. Environmental law enforcement departments have also made corresponding environmental punishment decisions based on local water pollutant emission standards, which prompts some pollutant enterprises to take advantage of the loopholes in more relaxed water pollutant emission standards in the YRDUA, so that they can emit water pollutants at a lower legal cost.

### 3.2.3. It Is Difficult to Eliminate Water Pollution Hidden in Different Area of Region Which Threatened Water Quality Safety in the YRDUA

Unification of the standards for the emission of water pollutants by the pharmaceutical industry in the YRDUA is difficult because the water pollution risks in this region have not been thoroughly investigated and dealt with. According to the statistics of Cn.com, there were a total of 1011 biopharmaceutical manufacturers in the YRDUA in 2019, 490 in Jiangsu Province, 105 in Shanghai Municipality, 223 in Zhejiang Province, and 193 in Anhui Province. The statistics regarding water pollutant emission types, concentration, scale, water pollution cost, and other data for more than 1000 biopharmaceutical enterprises have not been counted and shared among the regional environmental protection regulatory authorities. Since the central environmental protection supervision pilot in 2015, environmental pollution problems caused by biopharmaceutical enterprises have emerged one after another. For example, Zhejiang Haizheng Pharmaceutical was founded

by the Ministry of Environmental Protection to emit wastewater according to the COD standards, accumulating a high concentration of sewage in the cable trench in 2012 [22]. Anhui Shengda biopharmaceutical company was closed for renovations; according to the standard, this was due to excessive direct emissions through wastewater [23]. In 2017, the first round of environmental protection supervision officially began, and the environmental problems of four listed pharmaceutical companies in Jiangsu Province were exposed. Among them, the enterprises related to water pollutant emissions are the Jiangsu Huifeng Biological Agriculture Co., Ltd., which was found to have emitted wastewater under the table [24], Jiangsu Coffitte Biochemical Technology Co., Ltd., whose leachate always leaked and its biological drug preparation hazardous waste was mishandled, and so on. In 2018, Shanghai Magic Pharmaceutical Investment Management Co., Ltd. was also ordered to stop production by environmental protection authorities for emitting wastewater with pollutants from its subsidiary production base [25]. The abovementioned pharmaceutical manufacturing enterprises, which have been exposed due to water environmental pollution and subsequently rectified, gather in the Yangtze River Delta region, posing a potential threat to the environmental safety of inland rivers, lakes, and drinking water quality, which relates to human health.

## 4. Conclusions and Suggestions

### 4.1. Conclusions

This study aims to reveal differences in the standards for the emission of water pollutants and the risks and hidden dangers this poses for regional water environment safety. The findings can thus serve as a key influence for the industry in terms of integration and green transformation. A statistical and comparative analysis was used as the foundational method for the study, and a case study was also used to show the differences between the wastewater emission efficiency and pollutant emission standards of the pharmaceutical industry, thus proving that there are water environment problems in the YRDUA. The major conclusions are as follows:

The different urban pollutant emission standards have a significant impact on industry integration and green transformation; therefore, we have conducted a comparison of water pollutant emission standards in the three provinces and one municipality of the YRDUA. The main determinants influencing integration were found, namely, the race to the bottom of the biopharmaceutical industry, inconsistency in environmental protection regulation law, and trans-boundary water pollution risks.

It is recommended to use economic means of administration and marketing, a system guarantee mechanism, and an information sharing mechanism to realize the integration of water pollutant emission standards in the YRDUA. As we know, the integrated air pollutants emission standard in the YRDUA has been issued jointly by the three provinces and one municipality of this region on July 27th, 2021; however, there is still no integrated water pollutant emission standard in the YRDUA, so the suggestions of this research will promote the establishment of the integrated water pollutant emission standard in the YRDUA, and make an example for other regions' integrated environment standards across the country.

### 4.2. Suggestions

According to the results and discussion, and considering natural resource endowment, the water resources, water environment, and water ecology in the YRDUA have consistent characteristics; therefore, the integration of water pollutant emission standards in this region have inherent advantages regarding geographical location. Moreover, according to multi-stakeholder cooperative governance theory [26,27], suggestions to reflect (R) upon the integration of water pollutant emission standards are put forward.

### 4.2.1. Increase Government Funding for Local Water Pollution Governance and Encourage Industries to Adopt the Third-Party Governance Model for Pollution Control in the YRDUA

Relying on the Yangtze River Delta G60 scientific and technological corridor strategy and the G60 biopharmaceutical industry alliance, the strategy requires not only the development of a layout for the biopharmaceutical industry chain in the G60 scientific and technological corridor, in order to promote the healthy development of the regional biopharmaceutical industry through the unification of biopharmaceutical water pollutant emission standards in the YRDUA, but also to prevent the formation of a regional agglomeration of industrial space malformation with a low standard, but high emissions, by biopharmaceutical enterprises. Moreover, first, it is suggested that the local governments of the three provinces and one municipality increase financial investment in municipal water pollution governance and attract the whole biopharmaceutical industry chain with a balanced layout in local areas that have a higher governance level of urban water pollution. Second, it is suggested that the biopharmaceutical industrial park and enterprises in the YRDUA adopt the water pollutant emission concentration limits according to Jiangsu Province standard—with the strictest standards for emissions to be imposed upon the biopharmaceutical industry in this region—as the benchmark, so as to promote the third-party water pollution governance project of the biopharmaceutical industry in the YRDUA. Through the implementation of a strict standard for water pollutant emissions via the third-party governance model, the overall level of biopharmaceutical industry water pollution governance and water pollutant concentration limits of the standards in the YRDUA will be gradually increased; therefore, the integration of the biopharmaceutical water pollutant emission standards in the YRDUA will be achieved, and this experience can be used for demonstration purposes, and to promote to other industries in this region, with the integration of water pollutant emission standards in the YRDUA becoming gradually realized.

### 4.2.2. Unify Water Pollutant Emission Standards and Environmental Law Enforcement Standards in the YRDUA with a Mechanism Involving Shared Economic Responsibility

The first survey data of pollution sources of the three provinces and one municipality in the YRDUA show that about half of the enterprises of the biopharmaceutical industry in the region emitted wastewater to the drainage system of towns or industrial parks with terminal sewage treatment facilities. The wastewater emissions of these biopharmaceutical enterprises can effectively be supervised through online monitoring of the municipal sewage pipe network, and the remaining half of biopharmaceutical enterprises still emit wastewater directly into different environmental waters, which are in locations that are supervised and enforced in different administrative regions, and in accordance with those local water pollutant emission standards; hence, those biopharmaceutical enterprises can effectively perform water pollution prevention and governance in local areas. However, in the adjacent ecological environment space of the three provinces and one municipality, if the sewage emission enterprise implements the local water pollutant emission standard to be lower than that of others, it may be regarded as illegal behavior by the neighboring area (i.e., due to excess emissions). Polluters in the adjacent ecological environment space may even be investigated by environmental regulators in several neighboring regions at the same time, which not only increases the institutional cost of enterprises, but also causes a waste of resources in terms of law enforcement and case handling; therefore, it is suggested that local governments in the YRDUA adopt a mechanism of sharing economic responsibility, which is more efficient than the collective consultation system. Considering the cost of local water pollution governance and the impact on local economic development, the water pollutant emission standards of the five sub-industries of biopharmaceutical industry can be gradually unified. The environmental protection supervision and law enforcement standards for illegal emissions of industries in the YRDUA can then be unified by first unifying the industrial water pollutant emission standards.

4.2.3. Establish a Platform for Sharing Data and Governance Performance of Water Pollutant Emission in the YRDUA

The environmental protection supervision and law enforcement of biopharmaceutical polluting enterprises in the YRDUA should retain not only institutional innovation, but it should also require that technological innovation be supplemented by the implementation of policies and systems; therefore, in order to improve the efficiency of the joint prevention and governance mechanism of water pollution in the three provinces and one municipality of the YRDUA, it is suggested that the Economic and Information Technology Commission of the three provinces and one municipality, and the Ecological and Environmental Protection Bureau, jointly establish a statistical system of water pollutant emission data that is investigated by relevant governments and is provided by sewage enterprises in the YRDUA. Sharing real-time data for water pollutant emissions in the three provinces and one municipality of the region helps the environmental regulators in governance, and the biopharmaceutical companies to better control water pollutants in adjacent areas and check the potential water pollution risk of the biopharmaceutical industry in the YRDUA. It also helps environmental regulators ensure responsibility for water pollutant emissions and carry out the unified standard of environmental law enforcement more efficiently in terms of water pollution violations by the biopharmaceutical industry in this region. Moreover, based on the establishment of the statistical system of the water pollutant emissions of enterprises in the YRDUA, it is suggested that the Economic and Information Technology Commissions of the three provinces and one municipality, and the Ministry of Housing and Urban–Rural Development, jointly establish a performance sharing platform for water pollution governance based on a third-party governance mode, so as to control the level of water pollution governance performance in the three provinces and one municipality in this region. This would include the level of water pollution control technology, governance cost, treatment effect, and so on. This can also provide a technical and economic basis for formulating a unified water pollutant emission standard and environmental protection law enforcement standard for the biopharmaceutical industry in the YRDUA.

**Author Contributions:** Conceptualization, L.C.; methodology, L.C.; validation, X.L., S.Z. and M.L.; formal analysis, L.C.; investigation, M.L.; resources, L.C.; data curation, L.C.; writing—original draft preparation, L.C.; writing—review and editing, L.C.; visualization, M.L.; supervision, X.L.; project administration, L.C.; funding acquisition, M.L. All authors have read and agreed to the published version of the manuscript.

**Funding:** This research was funded by general project of Shanghai Social Science Foundation, project grant number 2021BGL015, its funder is Shanghai Office of Philosophy and Social Sciences, project name "The mechanism of environmental regulation on enterprises' green innovation under the background of carbon peak and carbon neutralization—From the perspective of policy perception"; funded by major project of Shanghai Academy of Social Science in 2022, project grant number 2022ZD021, its funder is Scientific Research Division of Shanghai Academy of Social Sciences, project name "Carbon Unlock Strategies and Pathways for Energy Development under Carbon Neutrality Vision".

**Institutional Review Board Statement:** Not applicable.

**Informed Consent Statement:** Not applicable.

**Data Availability Statement:** Not applicable.

**Conflicts of Interest:** The authors declare no conflict of interest. The funders had no role in the design of the study; in the collection, analyses, or interpretation of data; in the writing of the manuscript; or in the decision to publish the results.

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
