# Peer review of "Comparison of Regional Urban Water Pollutants Emission Standards and Determination of Factors Influencing Their Integration—A Case Study of the Biopharmaceutical Industry in the Yangtze River Delta Urban Agglomeration"

_sustainability, doi:10.3390/su14084741_

Round 1

Reviewer 1 Report

Dear Authors,

Hope that following suggestions will improve you paper quality and make it more compliant with the readers:

  1. Add literature review considering the topic of your research
  2. part of the paper wit results should include discussion also and omit suggestions 
  3. Suggestions should be placed in Conclusion and shortened - this will take the burden of reading from part 3 and make it more fluently.

Author Response

Dear Reviewer 1,

Thank you so much for reviewing my paper. And your comments are so sincere. According to each of your comments, I will response carefully and make a corresponding revision sufficiently in my paper as follow:

Comment 1: Add literature review considering the topic of your research

Response 1: According to the topic of my research, I add some literature review about environmental policy instruments like environmental standard studies in Section 1. Introduction Line37-42. And I have reordered the references.

Comment 2:part of the paper with results should include discussion also and omit suggestions.

Response 2: Yes, the results include discussion and suggestions, and the discussion is put forward in front of suggestion that’s why the part 3 looks like burden, so that I have adjusted the structure of paper, Section 3 is Results and Discussion; Section 4 is Conclusions and Suggestions.

Comment 3: Suggestions should be placed in Conclusion and shortened - this will take the burden of reading from part 3 and make it more fluently.

Response 3: As I have responded Comment 2, Suggestions is placed in Section 4 and shortened. Please check and review the modified version of my paper again.

Thank you again for your comments!

Your Sincerely,

Author, Dr Liping Cao

Reviewer 2 Report

The article is an analytical plan and is devoted to the environmental problems and sustainability of the development of regions.

My comments: The authors need to specify more clearly the theoretical (scientific) novelty of the research at the beginning of the article. Most of the information in the section "Materials and methods" p.2.1 refers to the section “Introduction”. In paragraph 2.2., the authors indicated the statistical analysis, but this method is not disclosed in detail in this Section. Since the work is carried out within the framework of the Grant, what is the main result the authors want to achieve with their research? In the Sections "Results" and "Discussions", what exactly do the authors propose to solve these environmental problems and how do they want to prove their position in the terms of different methods? The conclusions of the research should be reviewed: the authors should clearly indicate what purpose was in the study, what methods were used to achieve certain results of the work, what promising ways were proposed to solve the environmental problems under the study, the development forecasts and, ultimately, what contribution the authors' achievements will make to solving priority environmental problems of both the region and the country.

Author Response

Dear Reviewer 2,

Thank you so much for reviewing my paper. And your comments are so sincere. According to each of your comments, I will response carefully and make a corresponding revision sufficiently in my paper as follow:

Comment 1:The authors need to specify more clearly the theoretical (scientific) novelty of the research at the beginning of the article.

Response 1:In my paper, I have clearly stated the theoretical (scientific) novelty of the research. Please see Line 42-43 “this research wants to find the optimized path to decrease institutional cost by integrating the regional environment regulation”,and Line 103-107 “we take the water pollutants emission standards of biopharmaceutical industry in the YRDUA for example, and use statistical and comparative analysis methods to compare water pollutants emission standard in 4 jurisdictions within YRDUA to determine water environmental integration and coordination within the region; this analysis of comparation was never done in previous studies.”

Comment 2:Most of the information in the section "Materials and methods" p.2.1 refers to the section “Introduction”. In paragraph 2.2., the authors indicated the statistical analysis, but this method is not disclosed in detail in this Section.

Response 2:I have put the "Materials and methods" in “Introduction” section Line 103-105,and I have explained the statistical analysis method by using formula (1), see Line 165-167.

Comment 3:Since the work is carried out within the framework of the Grant, what is the main result the authors want to achieve with their research?

Response 3:As this paper is the research progress in funding Grant number 2021BGL015 “The mechanism of environmental regulation on enterprises' green innovation under the background of carbon peak and carbon neutralization—From the perspective of policy perception”, we want to prove the unified the environmental regulation in integrated region may improve the performance of enterprises' green innovation.

Comment 4:In the Sections "Results" and "Discussions", what exactly do the authors propose to solve these environmental problems and how do they want to prove their position in the terms of different methods?

Response 4:In order to decrease the burden of reading from Section 3 “Results and Suggestions” and make it more fluently, I have split Suggestions into Discussion part. In 3.2 Discussion, I have summarized the environmental problems those are competition to the bottom of the industry, environmental command and control regulations difficult to be consistent, difficult to eliminate water pollution hidden in different area of region. And we mainly use statistical analysis and case study method to prove the problems position. And suggestions are shortened put in Section 4.2.

Comment 5:The conclusions of the research should be reviewed: the authors should clearly indicate what purpose was in the study, what methods were used to achieve certain results of the work, what promising ways were proposed to solve the environmental problems under the study, the development forecasts and, ultimately, what contribution the authors' achievements will make to solving priority environmental problems of both the region and the country.

Response 5:Now Section 4 is Conclusions and Suggestions, in Section 4.1, the conclusions of the research have been reviewed, including the purpose in the study; the methods used; the promising ways proposed to solve the environmental problems under the study; the development forecasts and the contribution to solving priority environmental problems of both the region and the country. See Line 562-568 & 577-582.

Please check and review my paper again.

Thank you again for your comments, and happy new year!

Your Sincerely,

Author, Dr Liping Cao